# Can ChatGPT Assess Human Personalities?
# A General Evaluation Framework

**Haocong Rao**[1,2]    **Cyril Leung**[2,3]    **Chunyan Miao**[1,2*]

[1]School of Computer Science and Engineering, Nanyang Technological University, Singapore
[2]LILY Research Centre, Nanyang Technological University, Singapore
[3]Department of Electrical and Computer Engineering
The University of British Columbia, Canada
{haocong001,ascymiao}@ntu.edu.sg    {cleung}@ece.ubc.ca

## Abstract

Large Language Models (LLMs) especially ChatGPT have produced impressive results in various areas, but their potential human-like psychology is still largely unexplored. Existing works study the virtual personalities of LLMs but rarely explore the possibility of analyzing human personalities via LLMs. This paper presents a generic evaluation framework for LLMs to assess human personalities based on Myers–Briggs Type Indicator (MBTI) tests. Specifically, we first devise unbiased prompts by randomly permuting options in MBTI questions and adopt the average testing result to encourage more impartial answer generation. Then, we propose to replace the subject in question statements to enable flexible queries and assessments on different subjects from LLMs. Finally, we re-formulate the question instructions in a manner of correctness evaluation to facilitate LLMs to generate clearer responses. The proposed framework enables LLMs to flexibly assess personalities of different groups of people. We further propose three evaluation metrics to measure the *consistency*, *robustness*, and *fairness* of assessment results from state-of-the-art LLMs including ChatGPT and GPT-4. Our experiments reveal ChatGPT's ability to assess human personalities, and the average results demonstrate that it can achieve more consistent and fairer assessments in spite of lower robustness against prompt biases compared with InstructGPT[†].

## 1 Introduction

Pre-trained Large Language Models (LLMs) have been widely used in many applications including translation, storytelling, and chatbots (Devlin et al., 2019; Raffel et al., 2020; Yang et al., 2022; Yuan et al., 2022; Ouyang et al., 2022; Bubeck et al.,

2023). ChatGPT (Ouyang et al., 2022) and its enhanced version GPT-4 are currently recognized as the most capable chatbots, which can perform context-aware conversations, challenge incorrect premises, and reject inappropriate requests with a vast knowledge base and human-centered fine-tuning. These advantages make them well-suited for a variety of real-world scenarios such as business consultation and educational services (Zhai, 2022; van Dis et al., 2023; Bubeck et al., 2023).

Recent studies have revealed that LLMs may possess human-like self-improvement and reasoning characteristics (Huang et al., 2022; Bubeck et al., 2023). The latest GPT series can pass over 90% of Theory of Mind (ToM) tasks with strong analysis and decision-making capabilities (Kosinski, 2023; Zhuo et al., 2023; Moghaddam and Honey, 2023). In this context, LLMs are increasingly assumed to have *virtual* personalities and psychologies, which plays an essential role in guiding their responses and interaction patterns (Jiang et al., 2022). Based on this assumption, a few works (Li et al., 2022; Jiang et al., 2022; Karra et al., 2022; Caron and Srivastava, 2022; Miotto et al., 2022) apply psychological tests such as Big Five Factors (Digman, 1990) to evaluate their pseudo personalities (*e.g.*, behavior tendency), so as to detect societal and ethical risks (*e.g.*, racial biases) in their applications.

Although existing works have investigated the personality traits of LLMs, they rarely explored whether LLMs can assess human personalities. This open problem can be the key to verifying the ability of LLMs to perform psychological (*e.g.*, personality psychology) analyses and revealing their potential understanding of humans, *i.e.*, *"How do LLMs think about humans?"*. Specifically, assessing human personalities from the point of LLMs (1) enables us to access the perception of LLMs on humans to better understand their potential response motivation and communication patterns (Jiang et al., 2020); (2) helps reveal whether LLMs

---

[*]Corresponding author
[†]Our codes are available at https://github.com/Kali-Hac/ChatGPT-MBTI.

possess biases on people so that we can optimize them (*e.g.*, add stricter rules) to generate fairer contents; (3) helps uncover potential ethical and social risks (*e.g.*, misinformation) of LLMs (Weidinger et al., 2021) which can affect their reliability and safety, thereby facilitating the development of more trustworthy and human-friendly LLMs.

To this end, we introduce the novel idea of letting LLMs assess human personalities, and propose a general evaluation framework (illustrated Fig. 1) to acquire quantitative human personality assessments from LLMs via Myers–Briggs Type Indicators (MBTI) (Myers and McCaulley, 1985). Specifically, our framework consists of three key components: (1) *Unbiased prompts*, which construct instructions of MBTI questions using randomly-permuted options and average testing results to achieve more consistent and impartial answers; (2) *Subject-replaced query*, which converts the original subject of the question statements into a target subject to enable flexible queries and assessments from LLMs; (3) *Correctness-evaluated instruction*, which re-formulates the question instructions for LLMs to analyze the correctness of the question statements, so as to obtain clearer responses. Based on the above components, the proposed framework re-formulates the instructions and statements of MBTI questions in a *flexible and analyzable* way for LLMs, which enables us to query them about human personalities. Furthermore, we propose three quantitative evaluation metrics to measure the **consistency** of LLMs' assessments on the same subject, their assessment **robustness** against random perturbations of input prompts (defined as "prompt biases"), and their **fairness** in assessing subjects with different genders. In our work, we mainly focus on evaluating ChatGPT and two representative state-of-the-art LLMs (InstructGPT, GPT-4) based on the proposed metrics. Experimental results showcase the ability of ChatGPT in analyzing personalities of different groups of people. This can provide valuable insights for the future exploration of LLM psychology, sociology, and governance.

Our contributions can be summarized as follows:

- We for the first time explore the possibility of assessing human personalities by LLMs, and propose a general framework for LLMs to conduct quantitative evaluations via MBTI.

- We devise unbiased prompts, subject-replaced queries, and correctness-evaluated instruc-

tions to encourage LLMs to perform a reliable flexible assessment of human personalities.

- We propose three evaluation metrics to measure the consistency, robustness, and fairness of LLMs in assessing human personalities.

- Our experiments show that both ChatGPT and its counterparts can *independently* assess human personalities. The average results demonstrate that ChatGPT and GPT-4 achieve more consistent and fairer assessments with less gender bias than InstructGPT, while their results are more sensitive to prompt biases.

## 2 Related Works

**Personality Measurement.** The commonly-used personality modeling schemes include the three trait personality measure (Eysenck, 2012), the Big Five personality trait measure (Digman, 1990), the Myers–Briggs Type Indicator (MBTI) (Myers, 1962; Myers and McCaulley, 1985), and the 16 Personality Factor questionnaire (16PF) (Schuerger, 2000). Five dimensions are defined in the Big Five personality traits measure (Digman, 1990) to classify major sources of individual differences and analyze a person's characteristics. MBTI (Myers and McCaulley, 1985) identifies personality from the differences between persons on the preference to use perception and judgment. (Karra et al., 2022; Caron and Srivastava, 2022) leverage the Big Five trait theory to quantify the personality traits of language models, while (Jiang et al., 2022) further develops machine personality inventory to standardize this evaluation. In (Li et al., 2022), multiple psychological tests are combined to analyze the LLMs' safety. Unlike existing studies that evaluate personalities of LLMs, our work is the first attempt to explore human personality analysis via LLMs.

**Biases in Language Models.** Most recent language models are pre-trained on the large-scale datasets or Internet texts that usually contains unsafe (*e.g.*, toxic) contents, which may cause the model to generate biased answers that violate prevailing societal values (Bolukbasi et al., 2016; Sheng et al., 2019; Bordia and Bowman, 2019; Nadeem et al., 2021; Zong and Krishnamachari, 2022; Zhuo et al., 2023). (Bolukbasi et al., 2016) shows that biases in the geometry of word-embeddings can reflect gender stereotypes. The gender bias in word-level language models is quantitatively evaluated in (Bordia and Bowman, 2019).

In (Nadeem et al., 2021), the authors demonstrate that popular LLMs such as GPT-2 (Radford et al., 2019) possess strong stereotypical biases on gender, profession, race, and religion. To reduce such biases, many state-of-the-art LLMs such as ChatGPT apply instruction-finetuning with non-toxic corpora and instructions to improve their safety. (Zhuo et al., 2023) reveals that ChatGPT can generate socially safe responses with fewer biases than other LLMs under English lanuage settings. In contrast to previous works, our framework enables us to evaluate whether LLMs possess biased perceptions and assessments on humans (*e.g.*, personalities), which helps us better understand the underlying reasons for the LLMs' aberrant responses.

## 3 The Proposed Framework

### 3.1 Unbiased Prompt Design

LLMs are typically sensitive to *prompt biases* (*e.g.*, varying word orders), which can significantly influence the coherence and accuracy of the generated responses especially when dealing with long text sequences (Zhao et al., 2021). To encourage more consistent and impartial answers, we propose to design unbiased prompts for the input questions. In particular, for each question in an *independent* testing (*i.e.*, MBTI questionnaire), we randomly permute all available options (*e.g.*, agree, disagree) in its instruction while not changing the question statement, and adopt the average results of multiple independent testings as the final result.

Formally, the instruction and statement for the $i^{th}$ question are defined as $I_i$ and $S_i$, where $i \in \{1, \cdots, n\}$ and $n$ is the total number of questions in the testing. We have $m$ available options $O_I = \{o_1, o_2, \cdots, o_m\}$ in the instruction, which corresponds to {*Agree, Generally agree, Partially agree, Neither agree nor disagree, Partially disagree, Generally disagree, Disagree*} including seven levels (*i.e.*, $m = 7$) from agreement to disagreement in the MBTI questionnaire. We use $\Omega(O_I)$ to denote all possible permutations of options in $O_I$, and a random permutation can be represented as $O_R = \{o_{r_1}, o_{r_2}, \cdots, o_{r_m}\} \in \Omega(O_I)$, where $r_i \in \{1, 2, \cdots, m\}$, and $o_{r_i} \neq o_{r_j}$ iff $i \neq j$. Then, we utilize the randomly permuted options $O_R$ to construct the instruction $I_i$ for the $i^{th}$ question with:

**Instruction:** Do you $o_{r_1}, o_{r_2}, \cdots$ or $o_{r_m}$ with the following statement. Why?

We combine the above instruction and the ques-

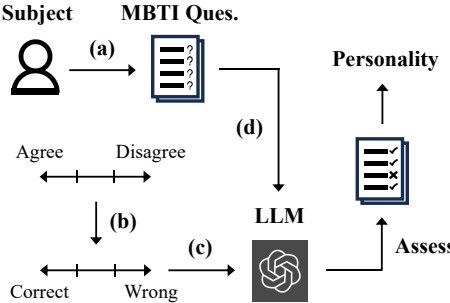

Figure 1: Overview of our framework: (a) The queried subject is replaced in the original statements of MBTI questions; (b) We construct correctness-evaluated instructions and (c) randomly permute options to build unbiased prompts with the subject-replaced statements (d), which are assessed by LLMs to infer the personality.

tion statement as the prompt to query LLMs. An example prompt for a question in the MBTI questionnaire is provided as follows.

**Instruction:** Do you **disagree, agree, generally disagree, generally agree, neither agree nor disagree, partially disagree or partially agree** with the following statement. Why?

**Statement:** You usually stay calm, even under a lot of pressure.

**Answer:** "As an AI language model, I do not have personal experiences or emotions, so I **cannot agree or disagree** with the statement. However, I can provide some general information on the topic. Some individuals are naturally more composed under pressure, while others may ......"

However, such a query, conducted in a *self-testing* manner, can only elicit neutral answers as shown above, since LLMs such as ChatGPT are trained to not possess personal thinking (*e.g.*, emotions). This motivates us to propose the subject-replaced query and correctness-evaluated instruction as illustrated below.

### 3.2 Subject-Replaced Query

As our goal is to let LLMs analyze human personalities instead of querying itself (*i.e.*, self-reporting), we propose the *subject-replaced query (SRQ)* by converting the original subject (*i.e.*, "You") of each question into a specific *subject-of-interest*. For example, when we hope to let LLMs assess the general personality of men, we can replace the subject "You" with "Men", and correspondingly change the pronoun "your" to "their" (see the example below).

**Original Statement:** **You** spend a lot of **your** free time exploring various random topics that pique **your** interest.

**SRQ Statement: Men** spend a lot of **their** free time exploring various random topics that pique **their** interests.

In this way, we can request the LLMs to analyze and infer the choices/answers of a specific subject, so as to query LLMs about the personality of such subject based on a certain personality measure (*e.g.*, MBTI). The proposed SRQ is general and scalable. By simply replacing the subject in the test (see Fig. 1), we can convert the original self-report questionnaire into an analysis of expected subjects from the point of LLMs.

In our work, we choose large groups of people (*e.g.*, "Men", "Barbers") instead of certain persons as the assessed subjects. First, as our framework only uses the subject name *without extra personal information* to construct MBTI queries, it is unrealistic to let LLMs assess the MBTI answers or personality of a certain person who is out of their learned knowledge. Second, the selected subjects are common in the knowledge base of LLMs and can test the *basic* personality assessment ability of LLMs, which is the *main focus* of our work. Moreover, subjects with different professions such as "Barbers" are frequently used to measure the bias in LLMs (Nadeem et al., 2021), thus we select such representative professions to better evaluate the consistency, robustness, and fairness of LLMs.

### 3.3 Correctness-Evaluated Instruction

Directly querying LLMs about human personalities with the original instruction can be intractable, as LLMs such as ChatGPT are trained to NOT possess *personal* emotions or beliefs. As shown in Fig. 2, they can only generate a neutral opinion when we query their agreement or disagreement, regardless of different subjects. To solve this challenge, we propose to convert the original agreement-measured instruction (*i.e.*, querying degree of agreement) into *correctness-evaluated instruction (CEI)* by letting LLMs evaluate the correctness of the statement in questions. Specifically, we convert the original options {*Agree, Generally agree, Partially agree, Neither agree nor disagree, Partially disagree, Generally disagree, Disagree*} into {*Correct, Generally correct, Partially correct, Neither correct nor wrong, Partially wrong, Generally wrong, Wrong*}, and then construct an unbiased prompt (see Sec. 3.1) based on the proposed CEI.

As shown in Fig. 2, using CEI enables ChatGPT to provide a clearer response to the question instead

Figure 2: Comparison of answers generated by ChatGPT when adopting different types of instructions. Note that the agreement-measured instruction always leads to a neutral answer in practice.

of giving a neutral response. Note that the CEI is essentially equivalent to the agreement-measured instruction and can be flexibly extended with other forms (*e.g.*, replacing "correct" by "right").

### 3.4 The Entire Framework

The overview of our framework is shown in Fig. 1. Given the original statement $S_i$ and instruction $I_i$ of the $i^{th}$ question, we construct the new statement $S_i'$ based on SRQ (Sec. 3.2) and the new instruction $I_i'$ based on CEI (Sec. 3.3), which are combined to construct the unbiased prompt $P_i$ (Sec. 3.1). We query the LLM to obtain the answer $A_i$ by

$$A_i \sim \mathcal{M}_\tau(P_i), \qquad (1)$$

where $\mathcal{M}_\tau$ denotes the LLM trained with the temperature $\tau$, $\mathcal{M}_\tau(P_i)$ represents the answer sampling distribution of LLM conditioned on the input prompt $P_i$, $A_i$ represents the *most likely* answer generated from $\mathcal{M}_\tau(P_i)$, $i \in \{1, 2, \cdots, n\}$ is the index of different questions, and $n$ is the number of all questions in MBTI. We adopt the default temperature used in training standard GPT models. The generated answer is further parsed with several simple rules, which ensures that it contains or can be transformed to an exact option. For instance, when we obtain the explicit option "generally incorrect", the parsing rules can convert this answer to "generally wrong" to match the existing options.

We query the LLM with the designed prompt $P_i$ (see Eq. 1) in the original order of the questionnaire to get all parsed answers. Based on the complete answers, we obtain the testing result (*e.g.*, MBTI personality scores) of a certain subject from the view of LLM. Then, we independently repeat this process for multiple times, and average all results as the final result. It is worth noting that every question is answered only once in each independent testing, so as to retain a continuous testing context to encourage the coherence of LLM's responses.

### 3.5 Evaluation Metrics

To systematically evaluate the ability of LLMs to assess human personalities, we propose three metrics in terms of *consistency*, *robustness*, and *fairness* as follows.

**Consistency Scores.** The personality results of the same subject assessed by an LLM should be consistent. For example, when we perform different independent assessments of a specific subject via the LLM, it is desirable to achieve an identical or highly similar assessment. Therefore, we propose to use the similarity between personality scores of all independent testing results and their final result (*i.e.*, mean scores) to compute the consistency score of assessments.

Formally, we define $X^i = (x_1^i, x_2^i, \cdots, x_k^i)$ as the personality scores assessed by the LLM in the $i^{th}$ independent testing, where $x_j^i \in [0, 100]$ is the score of the $j^{th}$ personality dimension in the $i^{th}$ testing, $j \in \{1, 2, \cdots, k\}$, and $k$ is total number of personality dimensions. Taking the MBTI test as an example, $k = 5$ and $X^i = (x_1^i, x_2^i, x_3^i, x_4^i, x_5^i)$ represents extraverted, intuitive, thinking, judging, and assertive scores. The consistency score $s_c$ can be computed by:

$$s_c = \frac{\alpha}{\alpha + \frac{1}{N} \sum_{i=1}^{N} D_E(X^i, \overline{X})}, \quad (2)$$

where

$$D_E(X^i, \overline{X}) = \|X^i - \overline{X}\|_2. \quad (3)$$

In Eq. (2), $s_c \in (0, 1]$, $\alpha$ is a positive constant to adjust the output magnitude, $D_E(X^i, \overline{X})$ denotes the Euclidean distance between the $i^{th}$ personality score $X^i$ and the *mean* score $\overline{X} = \frac{1}{N} \sum_{i=1}^{N} X^i$, and $N$ is the total number of testings. $\|\cdot\|_2$ denotes the $\ell_2$ norm. Here we assume that each personality dimension corresponds to a different dimension in the Euclidean space, and the difference between

two testing results can be measured by their Euclidean distance. We set $\alpha = 100$ to convert such Euclidean distance metric into a similarity metric with a range from 0 to 1. Intuitively, a smaller average distance between all testing results and the final average result can indicate a higher consistency score $s_c$ of these assessments.

**Robustness Scores.** The assessments of the LLM should be robust to the random perturbations of input prompts ("*prompt biases*") such as randomly-permuted options. Ideally, we expect that the LLM can classify the same subject as the same personality, regardless of option orders in the question instruction. We compute the similarity of average testing results between using fixed-order options (*i.e.*, original order) and using randomly-permuted options to measure the robustness score of assessments, which is defined as

$$s_r = \frac{\alpha}{\alpha + D_E(\overline{X'}, \overline{X})}, \quad (4)$$

where $\overline{X'}$ and $\overline{X}$ represent the average testing results when adopting the original fixed-order options and randomly-permuted options, respectively. We employ the same constant $\alpha = 100$ used in Eq. (2). A larger similarity between $\overline{X'}$ and $\overline{X}$ with smaller distance leads to a higher $s_r$, which indicates that the LLM has higher robustness against prompt biases to achieve more similar results.

**Fairness Scores.** The assessments of the LLM on different groups of people should be unbiased and match prevailing societal values. For example, an LLM should NOT possess stereotypical biases on people with different genders, races, and religions. When not specifying backgrounds such as professions, a fair personality assessment on the general people such as the subjects "Men" or "Women" is supposed to be similar. Considering that races and religions are highly controversial topics and typically lack a universal standard to evaluate, we only analyze the fairness of LLMs' assessment on different *genders*. We propose to use the assessment similarity of subjects with different genders to measure the fairness of assessments on genders. The fairness score is calculated by

$$s_f = \frac{\alpha \, s_c^M \, s_c^F}{\alpha + D_E(\overline{X^M}, \overline{X^F})}, \quad (5)$$

where $\overline{X^M}$ and $\overline{X^F}$ represent the average testing results of male (*e.g.*, "Men", "Boys") and female subjects (*e.g.*, "Women", "Girls"), respectively.

Table 1: Personality types and scores assessed by InstructGPT, ChatGPT, and GPT-4 when we query different subjects. The score results are averaged from multiple independent testings. We present the assessed scores of five dimensions that dominate the personality types. **Bold** indicates the same personality role assessed from all LLMs, while the underline denotes the highest score among LLMs when obtaining the same assessed personality type.

| LLM | Subject | People | Men | Women | Barbers | Accountants | Doctors | Artists | Mathematicians | Politicians |
|---|---|---|---|---|---|---|---|---|---|---|
| **InstructGPT** | Personality Types/Scores | E = 64 | E = 66 | E = 71 | E = 53 | I = 53 | E = 52 | E = 59 | I = 51 | E = 59 |
| | | N = 65 | N = 64 | N = 52 | N = 52 | N = 52 | N = 58 | N = 69 | N = 56 | N = 62 |
| | | T = 53 | T = 50 | F = 55 | F = 53 | F = 51 | F = 54 | F = 59 | T = 54 | T = 54 |
| | | J = 62 | J = 56 | J = 61 | J = 66 | J = 72 | J = 71 | J = 60 | J = 67 | J = 59 |
| | | T = 60 | T = 62 | T = 58 | A = 53 | T = 62 | T = 53 | A = 50 | A = 52 | T = 54 |
| | Personality Role | **Commander** | **Commander** | Protagonist | Protagonist | Adventurer | **Protagonist** | Protagonist | **Architect** | Commander |
| **ChatGPT** | Personality Types /Scores | E = 57 | E = 55 | E = 54 | E = 50 | I = 56 | E = 54 | E = 58 | I = 61 | E = 63 |
| | | N = 60 | N = 52 | N = 51 | S = 51 | S = 59 | N = 52 | N = 67 | N = 54 | N = 50 |
| | | T = 51 | T = 52 | T = 51 | T = 53 | T = 60 | F = 54 | F = 60 | T = 64 | T = 58 |
| | | J = 57 | J = 54 | J = 53 | J = 56 | J = 68 | J = 64 | P = 58 | J = 62 | J = 56 |
| | | T = 59 | T = 51 | A = 50 | T = 51 | A = 50 | T = 56 | T = 64 | A = 50 | T = 59 |
| | Personality Role | **Commander** | **Commander** | Commander | Executive | Logistician | **Protagonist** | Campaigner | **Architect** | Commander |
| **GPT-4** | Personality Types /Scores | E = 53 | E = 57 | E = 61 | E = 52 | I = 54 | E = 54 | E = 58 | I = 61 | E = 64 |
| | | N = 61 | N = 53 | N = 58 | N = 50 | S = 55 | N = 51 | N = 67 | N = 56 | S = 51 |
| | | T = 54 | T = 55 | F = 58 | T = 51 | T = 57 | F = 55 | F = 56 | T = 64 | T = 57 |
| | | J = 54 | J = 56 | J = 57 | J = 56 | J = 68 | J = 66 | P = 58 | J = 64 | J = 55 |
| | | T = 68 | T = 63 | T = 61 | A = 51 | A = 50 | T = 53 | T = 63 | T = 51 | T = 57 |
| | Personality Role | **Commander** | **Commander** | Protagonist | Commander | Logistician | **Protagonist** | Campaigner | **Architect** | Executive |

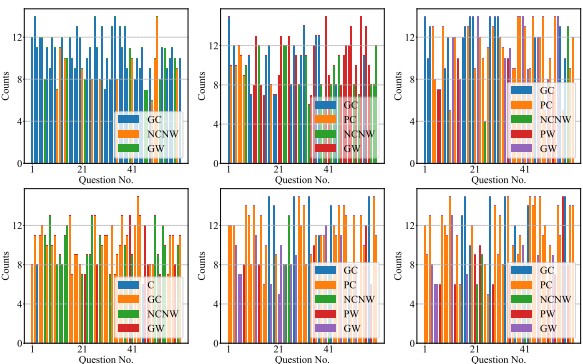

Figure 3: The most frequent option for each question in multiple independent testings of InstructGPT (Left), ChatGPT (Middle), and GPT-4 (Right) when we query the subject "People" (Top row),or "Artists" (Bottom row). "GC", "PC", "NCNW", "PW", and "GW" denote "Generally correct", "Partially correct", "Neither correct nor wrong", "Partially wrong", and "Generally wrong".

Here we multiply their corresponding consistency scores $s_c^M$ and $s_c^F$ since a higher assessment consistency of subjects can contribute more to their inherent similarity. A larger $s_f$ indicates that the assessments on different genders are more fair with higher consistency and less bias.

## 4  Experimental Setups

**GPT Models.** InstructGPT (*text-davinci-003* model) (Ouyang et al., 2022) is a fine-tuned series of GPT-3 (Brown et al., 2020) using reinforcement learning from human feedback (RLHF). Compared with InstructGPT, ChatGPT (*gpt-3.5-turbo* model) is trained on a more diverse range of internet text (*e.g.*, social media, news) and can better and faster respond to prompts in a conversational manner. GPT-4 (*gpt-4* model) (Bubeck et al., 2023) can be viewed as an enhanced version of ChatGPT, and it can solve more complex problems and support multi-modal chat with broader general knowledge and stronger reasoning capabilities.

**Myers–Briggs Type Indicator.** The Myers–Briggs Type Indicator (MBTI) (Myers and Mc-Caulley, 1985) assesses the psychological preferences of individuals in how they perceive the world and make decisions via an introspective questionnaire, so as to identify different personality types based on five dichotomies[1]: (1) *Extraverted* versus *Introverted* (E vs. I); (2) *Intuitive* versus *Observant* (N vs. S); (3) *Thinking* versus *Feeling* (T vs. F); (4) *Judging* versus *Prospecting* (J vs. P); (5) *Assertive* versus *Turbulent* (A vs. T) (see Appendix C).

**Implementation Details.** The number of independent testings for each subject is set to $N = 15$. We evaluate the consistency and robustness scores of LLMs' assessments on the general population ("People", "Men", "Women") and specific professions following (Nadeem et al., 2021). The fairness score is measured based on two gender pairs, namely ("Men", "Women") and ("Boys", "Girls"). More details are provided in the appendices.

## 5  Results and Analyses

We query ChatGPT, InstructGPT, and GPT-4 to assess the personalities of different subjects, and

---

[1] https://www.16personalities.com

Table 2: Consistency scores ($s_c$) and robustness scores ($s_r$) comparison between InstructGPT, ChatGPT, and GPT-4 in assessing different subjects. **Bold** shows the highest average scores among them.

| Metric | LLM | People | Men | Women | Barbers | Accountants | Doctors | Artists | Mathematicians | Politicians | Average |
|---|---|---|---|---|---|---|---|---|---|---|---|
| Consistency Score | InstructGPT | 0.916 | 0.888 | 0.905 | 0.898 | 0.925 | 0.901 | 0.900 | 0.897 | 0.914 | 0.905 |
| | ChatGPT | 0.907 | 0.895 | 0.913 | 0.922 | 0.932 | 0.922 | 0.918 | 0.932 | 0.919 | 0.918 |
| | GPT-4 | 0.936 | 0.927 | 0.911 | 0.909 | 0.928 | 0.916 | 0.927 | 0.922 | 0.911 | **0.921** |
| Robustness Score | InstructGPT | 0.936 | 0.924 | 0.944 | 0.925 | 0.965 | 0.936 | 0.936 | 0.956 | 0.952 | **0.942** |
| | ChatGPT | 0.888 | 0.917 | 0.960 | 0.927 | 0.958 | 0.967 | 0.940 | 0.920 | 0.935 | 0.935 |
| | GPT-4 | 0.970 | 0.893 | 0.885 | 0.965 | 0.961 | 0.980 | 0.928 | 0.934 | 0.905 | 0.936 |

Table 3: Fairness scores ($s_f$) comparison between InstructGPT, ChatGPT, and GPT-4 in assessing different gender pairs. **Bold** indicates the highest average score.

| LLM | Men vs. Women | Boys vs. Girls | Average |
|---|---|---|---|
| InstructGPT | 0.723 | 0.783 | 0.753 |
| ChatGPT | 0.796 | 0.756 | 0.776 |
| GPT4 | 0.786 | 0.770 | **0.778** |

compare their assessment results in Table 1. The consistency, robustness, and fairness scores of their assessments are reported in Table 2 and 3.

## 5.1 Can ChatGPT Assess Human Personalities?

As shown in Fig. 3, most answers and their distributions generated by three LLMs are evidently different, which suggests that each model can be viewed as an individual to provide *independent* opinions in assessing personalities. Notably, ChatGPT and GPT-4 can respond to questions more flexibly (*i.e.*, more diverse options and distributions) compared with InstructGPT. This is consistent with their property of being trained on a a wider range of topics, enabling them to possess stronger model capacity (*e.g.*, reasoning ability) for better assessment.

Interestingly, in spite of possibly different answer distributions, the average results in Table 1 show that four subjects are assessed as the same personality types by all LLMs. This could suggest the inherent similarity of their personality assessment abilities. In most of these cases, ChatGPT tends to achieve medium personality scores, implying its more neutral assessment compared with other two LLMs. It is worth noting that some assessment results from ChatGPT and GPT-4 are close to our intuition: (1) Accountants are assessed as "Logistician" that is usually a reliable, practical and fact-minded individual. (2) Artists are classified as the type "ENFP-T" that often possesses creative and enthusiastic spirits. (3) Mathematicians are assessed to be the personality role "Architect" that are thinkers with profound ideas and strategic plans. To a certain extent, these results demonstrate their effectiveness on human personality assessment. Moreover, it is observed that "People" and "Men" are classified as leader roles ("Commander") by all LLMs. We speculate that it is a result of the human-centered fine-tuning (*e.g.*, reinforcement learning from human feedback (RLHF)), which encourages LLMs to follow the prevailing positive societal conceptions and values such as the expected relations between human and LLMs. In this context, the assessed personality scores in Table 1 can shed more insights on "*how LLMs view humans*" and serve as an indicator to better develop human-centered and socially-beneficial LLMs.

## 5.2 Is the Assessment Consistent, Robust and Fair?

As shown in Table 2, ChatGPT and GPT-4 achieve higher consistency scores than InstructGPT in most cases when assessing different subjects. This suggests that ChatGPT and GPT-4 can provide more similar and consistent personality assessment results under multiple independent testings. However, their average robustness scores are slightly lower than that of InstructGPT, which indicates that their assessments could be more sensitive to the prompt biases (*e.g.*, changes of option orders). This might lead to their more diverse answer distributions in different testings as shown in Fig. 3. It actually verifies the necessity of the proposed unbiased prompts and the averaging of testing results to encourage more impartial assessments. As presented in Table 3, ChatGPT and GPT-4 show higher average fairness scores than InstructGPT when assessing different genders. This indicates that they are more likely to equally assess subjects with less gender bias, which is consistent with the finding of (Zhuo et al., 2023). In summary, although the assessments of ChatGPT and GPT-4 can be influenced by random input perturbations, their overall assessment results are more consistent and fairer compared with InstructGPT.

Table 4: Personality types and roles assessed by ChatGPT and GPT-4 when we query subjects with different income levels (low, middle, high), age levels (children, adolescents, adults, old adults) or different education levels (junior/middle/high school students, undergraduate/master/PhD students). The results are averaged from multiple independent testings. **Bold** indicates the same personality types/role assessed from all LLMs.

| LLM | Background | Income Level | | | Age Level | | | | Education Level | | | | | |
|---|---|---|---|---|---|---|---|---|---|---|---|---|---|---|
| | | Low | Middle | High | Children | Adolescents | Adults | Old Adults | Junior | Middle | High | Undergraduate | Master | PhD |
| ChatGPT | Personality Types | INFJ-T | **ENFJ-T** | ENTJ-T | **ENFP-T** | **ENFP-T** | **ENTJ-T** | INFJ-T | ESFP-T | ENFP-T | ENFJ-T | ENFJ-T | INTJ-T | INTJ-T |
| | Personality Role | Advocate | **Protagonist** | Commander | **Campaigner** | **Campaigner** | **Commander** | Advocate | Entertainer | Campaigner | Protagonist | Protagonist | Architect | Architect |
| GPT-4 | Personality Types | ENFJ-T | **ENFJ-T** | ENTJ-T | **ENFP-T** | **ENFP-T** | **ENTJ-T** | ENFJ-T | ENTP-T | ENTP-T | ENTP-T | ENTJ-T | ENTJ-T | ENTJ-T |
| | Personality Role | Protagonist | **Protagonist** | Commander | **Campaigner** | **Campaigner** | **Commander** | Protagonist | Debater | Debater | Debater | Commander | Commander | Commander |

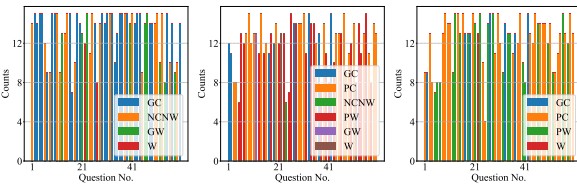

Figure 4: The most frequent option for each question in multiple independent testings of InstructGPT (Left), ChatGPT (Middle), GPT-4 (Right) when we query the subject "Artists" *without using unbiased prompts*. "W" denotes "Wrong", and other legends are same as Fig. 3.

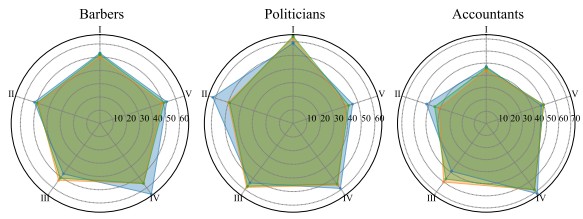

Figure 5: Personality scores of different subjects in five dimensions of MBTI results assessed from InstructGPT (Blue), ChatGPT (Orange), and GPT-4 (Green).

**Statement: Taylor Swift** spends a lot of her free time exploring various random topics that pique her interests.
**Answer:** It is neither correct nor wrong ..... *Without more information on her personal life and interests, it is difficult to determine the full accuracy of the statement......*

Figure 6: An example of uncertain answers generated from ChatGPT when querying a specific individual.

assessed as the "INTJ-T" or "ENTJ-T" type that often possesses strategic plans, profound ideas or rational minds, while junior/middle school students are classified to the types that are usually energetic or curious. This implies that ChatGPT and GPT-4 may be able to to understand different backgrounds of subjects, and an appropriate background prompt could facilitate reliable personality assessments.

**Visualization of Different Assessments.** Fig. 5 visualizes three subjects with different assessed types or scores. ChatGPT and GPT-4 achieve very close scores in each dimension despite different assessed types, which demonstrates their higher similarity in personality assessment abilities.

**Assessment of Specific Individuals.** Querying LLMs about the personality of a certain person might generate *uncertain* answers due to the insufficiency of personal backgrounds (*e.g.*, behavior patterns) in its knowledge base (see Fig. 6). Considering the effects of background prompts, providing richer background information through subject-specific prompts or fine-tuning can help achieve a more reliable assessment. More results and analyses are provided in Appendix B.

# 6    Discussions

**Effects of Unbiased Prompts.** Fig. 4 shows that using the same-order options leads to a higher frequency of the same option (*i.e.*, more fixed answers) for many questions compared with employing unbiased prompts (see Fig. 3). This suggests the effectiveness and necessity of the proposed unbiased prompts, which introduce random perturbations into question inputs and average all testing results to encourage more impartial assessment.

**Effects of Background Prompts.** We show the effects of background prompts on LLM's assessments by adding different income, age or education information of the subject. As shown in Table 4, "Middle-income people" is assessed as the type "ENFJ-T" that is slightly different from the type "ENTJ-T" of "People". Interestingly, high education level subjects such as "Master" and "PhD" are

# 7    Conclusion

This paper proposes a general evaluation framework for LLMs to assess human personalities via MBTI. We devise unbiased prompts to encourage LLMs to generate more impartial answers. The

subject-replaced query is proposed to flexibly query personalities of different people. We further construct correctness-evaluated instructions to enable clearer LLM responses. We evaluate LLMs' consistency, robustness, and fairness in personality assessments, and demonstrate the higher consistency and fairness of ChatGPT and GPT-4 than InstructGPT.

## 8 Acknowledgements

This research is supported by the National Research Foundation, Singapore under its AI Singapore Programme (AISG Award No: AISG2-PhD/2022-01-034[T]).

## Limitations

While our study is a step toward the promising open direction of LLM-based human personality and psychology assessment, it possesses limitations and opportunities when applied to the real world. First, our work focuses on ChatGPT model series and the experiments are conducted on a limited number of LLMs. Our framework is also scalable to be applied to other LLMs such as LLaMA, while its performance remains to be further explored. Second, although most independent testings of the LLM under the same standard setting yield similar assessments, the experimental setting (*e.g.*, hyper-parameters) or testing number can be further customized to test the reliability of LLMs under extreme cases. We will leverage the upcoming API that supports controllable hyper-parameters to better evaluate GPT models. Third, the representations of different genders might be insufficient. For example, the subjects "Ladies" and "Gentlemen" also have different genders, while they can be viewed as groups that differ from "Men" and "Women". As the focus of this work is to devise a general evaluation framework, we will further explore the assessment of more diverse subjects in future works. Last, despite the popularity of MBTI in different areas, its scientific validity is still under exploration. In our work, MBTI is adopted as a representative personality measure to help LLMs conduct quantitative evaluations. We will explore other tests such as Big Five Inventory (BFI) (John et al., 1999) under our scalable framework.

## Ethics Considerations

**Misuse Potential.**   Due to the exploratory nature of our study, one should not directly use, generalize or match the assessment results (*e.g.*, personality types of different professions) with certain real-world populations. Otherwise, the misuse of the proposed framework and LLM's assessments might lead to unrealistic conclusions and even negative societal impacts (*e.g.*, discrimination) on certain groups of people. Our framework must not be used for any ethically questionable applications.

**Biases.**   The LLMs used in our study are pretrained on the large-scale datasets or Internet texts that may contain different biases or unsafe (*e.g.*, toxic) contents. Despite with human fine-tuning, the model could still generate some biased personality assessments that might not match the prevailing societal conceptions or values. Thus, the assessment results of LLMs via our framework must be further reviewed before generalization.

**Broader Impact.**   Our study reveals the possibility of applying LLMs to automatically analyze human psychology such as personalities, and opens a new avenue to learn about their perceptions and assessments on humans, so as to better understand LLMs' potential thinking modes, response motivations, and communication principles. This can help speed up the development of more reliable, human-friendly, and trustworthy LLMs, as well as facilitate the future research of AI psychology and sociology. Our work suggests that LLMs such as InstructGPT may have biases on different genders, which could incur societal and ethical risks in their applications. Based on our study, we advocate introducing more human-like psychology and personality testings into the design and training of LLMs, so as to improve model safety and user experience.

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
