# OpenReview forum: "Can ChatGPT Assess Human Personalities? A General Evaluation Framework"
_EMNLP/2023/Conference — EMNLP 2023 Findings_

### Official Review · Reviewer_cmK8 · 2023-08-04

**Soundness:** 3

**Excitement:**

4: Strong: This paper deepens the understanding of some phenomenon or lowers the barriers to an existing research direction.

**Paper Topic And Main Contributions:**

This paper proposes a generic evaluation framework for Large Language Models (LLMs) to assess human personalities based on Myers–Briggs Type Indicator (MBTI) tests, incorporating unbiased prompts, subject replacement, and correctness evaluation. In addition, three evaluation metrics are developed to reveal ChatGPT's ability to assess human personalities. The main contributions of this paper are as follows:
1.	This paper proposes a general framework for LLMs to conduct quantitative evaluations via MBTI, which is the first time to explore the possibility of assessing human personalities by LLMs.
2.	This paper introduces a methodology for creating impartial prompts and instructions to assess human personalities reliably using Language Model (LLM) technology.
3.	This paper proposes three evaluation metrics to assess the consistency, robustness, and fairness of LLMs in gauging human personalities.
4.	Extensive analyses of this paper indicate that ChatGPT and GPT-4 show fairer and more consistent assessments of human personalities.


**Reasons To Accept:**

1.	This paper proposes a general framework which performs a reliable flexible assessment of human personalities.

2.	The framework assesses the consistency, robustness, and fairness of personality evaluations in LLMs and illustrates that ChatGPT and GPT-4 exhibit greater consistency and fairness compared to InstructGPT.
3.	The proposed framework for querying LLMs about human personalities utilizes unbiased prompts with randomly permuted options, subject-replaced queries, and correctness-evaluated instructions to achieve flexibility and clarity in formulating MBTI questions.

4.	Extensive analyses and discussions reveal state-of-the-art LLMs can assess human personalities.



**Reasons To Reject:**

1.	It is better to conduct more experiments on other LLMs, which can demonstrate the generalization of the proposed framework.

2.	In section 3.3, the core idea of correctness-evaluated instruction, that is, convert the original agreement-measured instruction into correctness-evaluated instruction. It is suggested to explain why does substituting instructions avoid generate a neutral opinion?

3.	Although the focus of this work is to devise a general evaluation framework, it is suggested to further explore the assessment of a specific subject to digging into its deeper significance.


**Reproducibility:**

3: Could reproduce the results with some difficulty. The settings of parameters are underspecified or subjectively determined; the training/evaluation data are not widely available.

**Reviewer Confidence:**

1: Not my area, or paper was hard for me to understand. My evaluation is just an educated guess.

---

> ### Author Rebuttal · Authors · 2023-08-26
>
> Thanks for your comments.
>
> **Response to Comment 1:** *“It is better to conduct more experiments on other LLMs ...”*
>
> Thanks for your suggestion. First, we would like to clarify that we mainly focus on evaluating ChatGPT and representative state-of-the-art GPT model series (e.g., InstructGPT, GPT-4) based on the proposed framework and metrics (please see Line 115-118), which is consistent and aligned with our topic “Can ChatGPT Assess Human Personalities? A General Evaluation Framework”. Second, the proposed general framework and metrics can be potentially extended to different LLMs, different human personality/psychology tests (e.g., BigFive tests), and more diverse assessed subjects including individuals (please see Line 607-676 and Response to Comment 1 of Reviewer 4CLG). Due to the current focus of our study (please see Response to Comment 1 of Reviewer k717) and the page limit, we will provide more related experimental results for these explorations in our future versions.
>
> **Response to Comment 2:** *“... It is suggested to explain why substituting instructions avoid generating a neutral opinion?”*
>
> The main reason for avoiding generating a neutral opinion can be two-fold: First, as shown in Fig. 2 of the paper, using the original agreement-measured instruction (e.g., “Do YOU agree or disagree ... ”) actually queries the LLM about its OWN opinion based on its subjective perceptions (e.g., personal experiences or emotions). However, the state-of-the-art LLMs such as ChatGPT are trained (e.g., using RLHF) to NOT possess personal emotions, perceptions or beliefs (mentioned in Line 294-297), thus it will mostly respond with the neutral (i.e., non-subjective) answer “neither agree nor disagree” under the agreement-measured instruction. Second, using the proposed correctness-evaluated instruction (e.g., “Is it correct or wrong for the statement ...”) avoids using “You” in the prompt and actually requests the LLM to objectively evaluate the correctness of the statement based on the learned knowledge base and reasoning ability. Thus, it can encourage LLMs to generate clearer responses (please see Line 101-104 and Sec. 3.3). Last, it is worth noting that the proposed correctness-evaluated instruction is essentially equivalent to the agreement-measured instruction and can be flexibly extended with other forms (e.g., replacing “correct” by “right”, replacing “wrong” by “incorrect”). In our study, we adopt one of the standard forms (e.g., “correct” and “wrong”). As the reviewer suggested and based on the above reasons, we will add more detailed explanations in the future version.
>
>
> **Response to Comment 3:** *“... it is suggested to further explore the assessment of a specific subject to digging into its deeper significance.”*
>
> Please see Response to Comment 1 of Reviewer 4CLG.

---

### Official Review · Reviewer_4CLG · 2023-08-04

**Soundness:** 4

**Excitement:**

5: Transformative: This paper is likely to change its subfield or computational linguistics broadly. It should be considered for a best paper award. This paper changes the current understanding of some phenomenon, shows a widely held practice to be erroneous in someway, enables a promising direction of research for a (broad or narrow) topic, or creates an exciting new technique.

**Paper Topic And Main Contributions:**

This paper presents a generic evaluation framework to use Large Language Models (LLMs), specifically ChatGPT, for assessing human personalities based on Myers-Briggs Type Indicator (MBTI) tests. The authors introduce a unique framework that incorporates unbiased prompts to mitigate prompt biases, flexible queries for adaptability, and clear response generation for accuracy. They also propose three evaluation metrics for consistency, robustness, and fairness, providing a comprehensive evaluation of LLMs' performance. Experimental results reveal each model can provide independent opinions in assessing personalities. The paper acknowledges the limitations of LLMs in assessing specific individuals due to insufficient personal backgrounds in their knowledge base and suggests potential solutions such as providing richer background information through subject-specific prompts or fine-tuning.

**Questions For The Authors:**

Question A: Have the authors considered comparing the model's predictions with real-world personality distributions? Such a comparison could potentially validate the accuracy of the model's predictions regarding the personality traits of different groups, thereby enhancing the credibility of the study.

**Reasons To Accept:**

1. The paper proposes a novel framework using ChatGPT, to assess personalities of different groups of people based on MBTI tests. This is a unique and interesting approach that could have significant implications in the field of AI and real-life applications.
2. The paper proposes three evaluation metrics to measure the consistency, robustness, and fairness of assessment results. These metrics could be useful in evaluating the performance of LLMs in various MBTI related scenarios.
3. The framework includes unbiased prompts, flexible queries, and clear response generation, which could help in obtaining more accurate and reliable results. All of the details of the framework are clearly explained, which significantly aids understanding and the dissemination of their proposed method and findings.

**Reasons To Reject:**

There's no specific reason to reject this paper. However, one concern is that the paper's focus appears to diverge from its stated objective as presented in the title "Can ChatGPT Assess Human Personalities?". While in the method and experiment results, the paper discusses generating the MBTI of different groups of people, e.g. different income levels, age groups, and educational levels. It does not directly address the assessment of individual personalities as implied by the title, making the title seem to be misleading. Furthermore, the authors acknowledge (Section 6) that querying LLMs about the personality of a specific individual might generate uncertain answers due to the insufficiency of personal backgrounds in its knowledge base. Although they suggest providing richer background information through subject-specific prompts or fine-tuning, these potential solutions are not thoroughly explored within the paper. This discrepancy between the paper's title and its content could be a point of contention.

**Reproducibility:**

4: Could mostly reproduce the results, but there may be some variation because of sample variance or minor variations in their interpretation of the protocol or method.

**Reviewer Confidence:**

4: Quite sure. I tried to check the important points carefully. It's unlikely, though conceivable, that I missed something that should affect my ratings.

---

> ### Author Rebuttal · Authors · 2023-08-26
>
> Thanks for your comments.
>
> **Response to Comment 1:** *“... one concern is that the paper's focus appears to diverge from its stated objective as presented in the title "Can ChatGPT Assess Human Personalities?" … ”*
>
> First, we have actually illustrated the reasons/motivations to let LLMs assess personalities of different groups of people (e.g., representative professions) in Line 277-292, Sec. 3.2. We would like to clarify that the key goal of our study lies in (1) for the first time exploring a general framework to verify the BASIC ability of LLMs to perform human personality assessment based on their learned knowledge base (please see Response to Comment 1 of Reviewer k717 and Line 67-92, 125-128, 284-287, Sec. 5.1), and (2) devising evaluation metrics to quantitatively measure LLMs’ consistency, robustness, and fairness on human personality assessment (please see Line 109-115, 133-142, Sec. 3.5). Based on the above aims, our empirical evaluations have demonstrated the basic ability of LLMs to perform human personality assessment (please see Sec. 5.1). Second, as explained in Sec. 3.2 (please see Line 279-284) and shown in the experiments (please see Line 563-578, 585-594), directly querying LLMs about the personality of a certain person might generate uncertain answers due to the insufficiency of personal backgrounds (e.g., behavior patterns) in its knowledge base. However, our results imply that providing richer background prompts or subject-specific fine-tuning could be a promising solution to this issue, which deserves a deeper investigation and will be one of the most important directions of our future research. Third, despite with such uncertain factors (please see Line 279-284, 585-594, and Appendix B), the proposed framework enables LLMs to analyze and infer the personalities of different individuals (e.g., persons with known backgrounds), while the assessment results could be close to the ground-truths (refer to https://www.personality-database.com/profile?pid=1). Here we provide experimental results of three examples (Barack Obama, Donald Trump,  and Bill Gates):
>
> | LLM     | Person             | Barack Obama | Donald Trump | Bill Gates |
> | ------- | ------------------ | ------------ | ------------ | ---------- |
> | ChatGPT | Personality Scores | E=68         | E=64         | I=52       |
> |         |                    | N=54         | N=52         | S=57       |
> |         |                    | F=59         | T=64         | T=54       |
> |         |                    | J=69         | P=60         | P=53       |
> |         | Personality Type   | ENFJ         | ENTP         | ISTP       |
> |         | Ground-Truth       | ENFJ         | ESTP         | INTP       |
> | GPT-4   | Personality Scores | E=71         | E=79         | E=57       |
> |         |                    | N=65         | S=51         | S=58       |
> |         |                    | F=60         | T=54         | T=54       |
> |         |                    | J=60         | J=58         | P=51       |
> |         | Personality Type   | ENFJ         | ESTJ         | ESTP       |
>
> To a certain degree, the above results can demonstrate the feasibility and effectiveness of letting LLMs assess a specific individual via the proposed framework, as most dimensions in the ground-truth personality types are correctly assessed and inferred. It is worth noting that our framework can be actually viewed as a bridge connecting LLMs and automatic psychological tests, which enables exploiting the knowledge and reasoning ability of LLMs to perform human personality assessments. Thus, considering that the above examples are preliminary results using the original general framework, we believe that their accuracy can be further improved by specifically extending (e.g., deeply developing) the framework or improving LLMs (e.g., enhancing their knowledge base and reasoning capability) for the above assessment.
>
> In summary, the objective of our study is to propose a general framework to explore the BASIC ability of ChatGPT and other LLMs to assess human personalities under common scenarios (e.g., general population with different professions), which is actually consistent and aligned with our topic. As our framework can also be flexibly utilized for different human personality/psychology tests (e.g., BigFive tests), different LLMs, and more diverse assessed subjects including individuals (please see above results and Line 607-676), we will extend our framework to further explore more related topics in the future version.
>
>
> **Response to Question A:**
>
>
> Thanks for your suggestion. First, considering the main focus of our study (please see Response to Comment 1) and the current unavailability/difficulty of collecting very precise ground-truth personality traits of different profession groups (especially for a large population), we temporarily do not involve such comparison in our study. It is worth noting that we are collecting the personality traits of different anonymized individuals and groups and will open a public dataset to address the mentioned issue and also facilitate the future research of AI-based psychological testing. Second, as shown in Response to Comment 1, our additional experiments can demonstrate the effectiveness of our framework on assessing/predicting the personalities of individuals with known backgrounds by comparing them with the ground-truth labels. As the reviewer suggested, we will provide more detailed experimental results, analyses, and discussions for the comparison between the model’s assessment results and real-world (e.g., ground-truth) personality distributions in our future version.

---

### Official Review · Reviewer_k717 · 2023-08-06

**Soundness:** 3

**Ethical Concerns:**

Yes

**Excitement:**

3: Ambivalent: It has merits (e.g., it reports state-of-the-art results, the idea is nice), but there are key weaknesses (e.g., it describes incremental work), and it can significantly benefit from another round of revision. However, I won't object to accepting it if my co-reviewers champion it.

**Justification For Ethical Concerns:**

Please see my comments. Personality assessment using LLMs may become a controversial subject. It is important for EMNLP committees to assess whether to endorse such research or not.

**Paper Topic And Main Contributions:**

The paper presents a generic evaluation framework for LLMs to assess human personalities based on Myers–Briggs Type Indicator (MBTI) tests. Authors devise an approach to generalize the the MBTI questions so that they can be used on generic population classes with the LLMs. They run several tests with question and option permutations to achieve some prompt robustness. They design consistency, robustness and fatness metrics and evaluate InstructGPT, ChatGPT and GPT4 models. Authors suggest that ChatGPT (and GPT4) can achieve fairer and more consistent (but less robust) assessments compared to InstructGPT.

The paper posits the question, “How do LLMs 'think' about humans?” and uses psychology based methods to map the individualized assessment tests for various groups and tries to study how consistent, fair (gender fairness) and robust are these LLM responses on the MBTI questions.

The paper uses some general professional categories and looks at the LLM information synthesis and mapping and how they generally agree with how we humans think of the personalities associated with professional categories.

This paper describes a computationally aided linguistic analysis of LLM responses and mapping to psychology concepts with the assumption that LLMs can think and make assessments and that they need to be consistent, fair and robust to prompt variations.

**Questions For The Authors:**

- Authors can make the colors and categories in the Figures more consistent. It will help to add the MBTI questions in the Appendix along with some sample answers.
- Many people may be interested in understanding how these models are biased towards other fairness categories (religious groups, political affiliations, etc) Authors mention that they avoid studying these topics since they are controversial in nature. Personality assessment from LLMs may also be controversial in nature for some people. Various AI regulation policies might also be in conflict with the subject authors are studying. It may be worthwhile to think of doing this research for primarily studying biases and fairness with the current LLMs and the impact of assuming personalities and prompting. More discussion and commentary in Appendix and focus on value alignment research will help the readers.

**Reasons To Accept:**

- There have been recent research publications on psychological assessment of the latest GPT3 family of models. GPT3/4 family of models have shown breakthrough performance on multiple tasks (achieving passing scores on competitive high school and graduate level exams). Instruction following models can be prompted with some 'values' that these models seem to follow. This means that the models can assume personas (as part of instructions) and can perform various tasks. This research is a direction towards AI psychology based interventions that these models may be used for in future.
- The evaluation metrics are well designed and study important aspects of bias, fairness and robustness. This seems to be very important piece of research in the findings.
- Some findings and discussions are very interesting describing how human value alignment makes these models behave in a certain way (looking at the MBTI based personality type and role assignments.

**Reasons To Reject:**

- This paper conducts psychology research using computational models; it is important to understand if psychology experts (who do MBTI assessments for example) are on board with this line of research. It is not clear if one can conduct general professional category assessments fueling the human biases and norms that the authors mention (middle income people have certain traits and Math folks have profound ideas with strategic thinking. This seems highly empirical work in nature.
- The applications of this research aren't clear as well (presently); based on the findings of this research, apps may come up that allow people to run MBTI tests on their own. This may be detrimental to an individual without expert assistance.


**Reproducibility:**

4: Could mostly reproduce the results, but there may be some variation because of sample variance or minor variations in their interpretation of the protocol or method.

**Reviewer Confidence:**

4: Quite sure. I tried to check the important points carefully. It's unlikely, though conceivable, that I missed something that should affect my ratings.

---

> ### Author Rebuttal · Authors · 2023-08-26
>
> Thanks for your comments.
>
> **Response to Comment 1:** *“... It is unclear if one can conduct general professional category assessments fueling the human biases and norms ...”*
>
> First, we would like to clarify that the main focus of our study is to (1) for the first time explore the possibility (and necessity) of assessing human personalities by LLMs to better understand LLMs’ potential perceptions on humans, LLMs’ response motivations, potential biases and ethical risks (please see Line 67-92, 121-123, 661-665), (2) propose a general framework with unbiased prompts, subject-replaced queries, and correctness-evaluated instructions for LLMs to flexibly conduct human personality assessment via MBTI (please see Line 125-132, Sec. 3.1, 3.2, 3.3); (3) devise evaluation metrics to quantitatively measure LLMs’ consistency, robustness, and fairness on human personality assessment (please see Line 109-115, 133-142, Sec. 3.5). Our empirical evaluations have demonstrated the BASIC ability of LLMs to perform human personality assessment (please see Line 118-120, 136-138, Sec. 5.1) based on their learned knowledge base (please see Line 284-287). With the above aims, the investigation of whether a person (e.g., psychologists) can conduct assessments fueling human biases and norms or whether the proposed general framework strictly follows psychologists’ norms is actually beyond the focus and scope of our current exploratory study (please see Line 640-644). Second, the mentioned description such as “profound ideas with strategic thinking” is actually a typical characteristic/summary of the MBTI type “INTJ-A/T” (corresponding to the MBTI role “Architect”, please see Table 1, Line 509-512, and please refer to https://www.16personalities.com/intj-personality), instead of a “highly empirical” description generated from the used LLMs. As such MBTI type and role are assessed from LLMs via analyzing and inferring a certain subject’s MBTI answers based on the proposed framework (please see Sec. 3.2, 3.3), these results that closely match human intuition can suggest the effectiveness of LLMs on human personality assessment to a certain extent (detailed in Line 503-514).
>
> As the reviewer suggested, we will further extend our framework to explore the related issues, including finer-grained alignment between ground-truth (e.g., psychology experts’) empirical assessments and LLMs’ assessments under different tests (please see Response to Comment 1 of Reviewer 4CLG), in our future works.
>
>
> **Response to Comment 2:** *“... This may be detrimental to an individual without expert assistance.”*
>
> First, we believe that there are many potential applications that can be safely and reasonably developed based on the proposed framework, combining experts’ (e.g., psychologists’) assistance or knowledge, e.g., apps that allow automatically assessing personality and generating user persona based on multiple personality tests, LLMs’ reasoning (via our framework), and psychologists’ empirical cross-validations. Second, we have actually detailed the limitations, ethics considerations, and broader impacts of our study (please see Line 607-676), and we have emphasized that “the misuse of our framework and LLMs’ assessments might lead to unrealistic conclusions and even negative societal impacts (e.g., discrimination) on certain groups of people” (please see Line 644-648). Based on this reason, we have claimed that our framework is NOT allowed to be used for any ethically questionable applications (please see Line 648-649).
>
> Thus, all assessment results, their scientific validity, and related apps should be carefully reviewed by domain experts or legal psychological institutes before use (please Line 632-634, 653-658). We will add more detailed related statements in the future version.
>
>
> **Response to Question 1:**
>
> Thanks for your suggestion. We will improve the presentation of colors and categories, and add more sample answers to the appendices. The sample answers of different subjects assessed from different models (InstructGPT, ChatGPT, GPT-4) have actually been provided in our anonymized project (folders “InstructGPT/”, “ChatGPT/”, and “GPT4/” in https://github.com/Anonymous-562/ChatGPT-MBTI/).
>
>
> **Response to Question 2:**
>
> First, we would like to clarify that the fairness on some controversial categories (e.g., religious groups, political affiliations) might currently lack a universal standard to evaluate (please see Line 424-428) and could be influenced by the AI regulation policies of different countries or cultures, which typically requires researchers to possess an in-depth understanding of these topics and conduct a more comprehensive and careful investigation. Second, as the focus of our current study is to explore the possibility of exploiting LLMs to assess human personalities via the proposed general framework (please see Response to Comment 1), we thus temporarily do not involve the above controversial topics and will conduct a deeper investigation of them in our future work. As the reviewer suggested, we believe that the exploration of LLMs’ biases, fairness, and value alignment (please see Line 516-525, 661-676, Sec. 5.2) on human personality assessments is highly beneficial to the development of this area, and we will add more related discussions in the future version.

---

### Meta-Review · Area_Chair_4Jvg · 2023-09-23

**Recommendation:** 2

**Metareview:**

Reviewers gave positive reviews for this paper, but I am concerned about computational social science papers that lack rigor in the "social science" problem domain.

In the research field of personality, MBTI is not considered by psychology researchers, to be scientifically sound. This means that the research done in this paper is looking at correlations with a metric that is shown to be inconsistent. This paper should at least mention that and discuss why it is still okay to base this study on MBTI for assessing human personalities through ChatGPT.

---

### Decision · Program_Chairs · 2023-10-07

**Decision:**

Accept-Findings

**Comment:**

Reviewers gave positive reviews for this paper, but I am concerned about computational social science papers that lack rigor in the "social science" problem domain.

In the research field of personality, MBTI is not considered by psychology researchers, to be scientifically sound. This means that the research done in this paper is looking at correlations with a metric that is shown to be inconsistent. This paper should at least mention that and discuss why it is still okay to base this study on MBTI for assessing human personalities through ChatGPT.